# A Mixture of Exogenous Emulsifiers Increased the Acceptance of Broilers to Low Energy Diets: Growth Performance, Blood Chemistry, and Fatty Acids Traits

**DOI:** 10.3390/ani10030437

**Published:** 2020-03-05

**Authors:** Ahmed A. Saleh, Khairy A. Amber, Mahmoud M. Mousa, Ahmed L. Nada, Wael Awad, Mahmoud A.O. Dawood, Abd El-Moneim E. Abd El-Moneim, Tarek A. Ebeid, Mohamed M. Abdel-Daim

**Affiliations:** 1Department of Poultry Production, Faculty of Agriculture, Kafrelsheikh University, Kafr El-Sheikh 33516, Egypt; a_saleh2006@yahoo.com (A.A.S.); Khairy.amber@agr.kfs.edu.eg (K.A.A.); kareemmousa83@yahoo.com (M.M.M.); ahmed.lotfi@orkila.com (A.L.N.); tarkamin@gmail.com (T.A.E.); 2Animals Production Institute, Agriculture Research Center, Ministry of Agriculture, Giza 12651, Egypt; wawad74@yahoo.com; 3Department of Animal Production, Faculty of Agriculture, Kafrelsheikh University, Kafrelsheikh 33516, Egypt; 4Biological Application Department, Nuclear Research Center, Atomic Energy Authority, Abu-Zaabal 13759, Egypt; aeabdelmoneim@gmail.com; 5Department of Zoology, College of Science, King Saud University, P.O. Box 2455, Riyadh 11451, Saudi Arabia; abdeldaim.m@vet.suez.edu.eg; 6Pharmacology Department, Faculty of Veterinary Medicine, Suez Canal University, Ismailia 41522, Egypt

**Keywords:** broilers, blood chemistry, emulsifiers, growth, nutrient utilization, TBARs

## Abstract

**Simple Summary:**

Since fat energy is cheaper than carbohydrate energy, it is profitable to increase fat content in broiler diets. One of the factors that limits using high levels of fat in broiler diets is the indigestion of fat, because bile secretion in broilers is not efficient in the first days of age. In this sense, using exogenous emulsifiers in the high-fat diet enhanced fat utilization and digestive metabolism. In the current study, birds fed the basal diet and another two low-energy diets (−50 kcal/kg than control) with or without emulsifiers (500 g/ton). The obtained results revealed that the emulsifier’s supplementation to low-energy diets enhanced fat utilization and resulted in positive effects on growth performance, nutrients utilization, lipid peroxidation, and modified plasma lipid profiles in broilers.

**Abstract:**

To investigate the influence of emulsifiers on broilers fed low-energy diets, the birds were distributed into three sets—the control was fed the basal diet, the second group was fed diets 50 kcal/kg less than control, and the third group was fed diets 50 kcal/kg less than control and supplemented with 500 g/ton of emulsifiers. The used mixture of exogenous emulsifiers contains phosphatidyl choline, lysophosphatidyl choline, and polyethylene glycol ricinoleate. Although the feed intake was not meaningfully affected by dietary low-energy level with emulsifier inclusion (*P* = 0.42), the weight gain and FCR were clearly enhanced (*P* = 0.005 and *P* = 0.044, respectively). Protein and lipids utilization were decreased by reducing energy level, but they were increased by emulsifier supplementation (*P* = 0.022 and *P* = 0.011, respectively). Liver thiobarbituric acid-reactive substances (TBARs) and muscle palmitic acid concentrations were decreased by reducing the energy level and emulsifier’s supplementation (*P* = 0.014 and *P* = 0.042, respectively). However, muscle total lipids and α-tocopherol, oleic acid, linoleic acid, and α-linolenic acid were not affected by dietary treatments (*P* > 0.05). Interestingly, the plasma total cholesterol, HDL-cholesterol, total protein, and globulin were decreased in the low-energy group without emulsifier but they were increased by emulsifier supplementation (*P* = 0.008, *P* = 0.005, *P* = 0.037, and *P* = 0.005, respectively). It could be concluded that the mixture of emulsifier supplementation to low-energy diets enhanced fat utilization and resulted in positive effects on the growth performance, nutrient utilization, lipid peroxidation, and modified plasma lipid profiles in broilers. Getting such benefits in broilers is a necessity to reduce the feed cost and consequently the price of the product, which will lead to improved welfare of mankind.

## 1. Introduction

Fats and oils are the most essential energy sources in broilers’ diets as a worthy way for gathering the high energy demands for the highest growth rates of broiler chickens [1,2]. In addition, the evident merits of high-caloric-density lipids result in an excellent caloric impact [3,4]. The failure of the broilers to gain lipids is assigned to bad emulsification rather than the shortage in lipase secretion, which led to great interest in the possibility of using exogenous emulsifiers to improve the utilization of lipids in broiler chickens [5]. Several previous studies indicated that supplementation with bile acids or bile salts improve the utilization of dietary fat by chicks because of limited endogenous secretion [6,7]. 

Big amounts of fat are used in broilers feed, especially strains that require high-energy diets [8]. In various situations, the actual utilization of lipids is specified by the cost linkage between energy content and yellow corn energy content [9]. Since fat energy is cheaper than carbohydrate energy, it is profitable to increase fat content in broiler diets. One of the factors that limits using high levels of fat in broiler diets is the indigestion of fat, because bile secretion in broilers is not efficient in the first days of age [10]. In this time, using exogenous emulsifiers in the high-fat diet enhances fat utilization and digestive metabolism [3,5]. Moreover, San Tan et al. [11] recommended that supplementation of emulsifiers in the early stages of age improved digestion and absorption of the fats and enhanced growth performance in broilers. Although, supplemented bile acids (including cholic acid and chenodeoxycholic acid) and bile salts (taurocholate) improved the absorption of fat in broilers [10]. However, supplemented bile acid in the diets was not economically applicable due to the high cost. Thus, emulsifiers might be included in broilers’ diets to reduce the surface tension of water [5]. Bontempo et al. [12] illustrated that emulsifier supplementation in broiler diets consisting of a vegetal bidistilled oleic acid and glycerol polyethylene glycol ricinoleate had a positive effect on growth performance, feed efficiency, carcass dressing, and plasma lipid metabolism. Furthermore, Siyal et al. [5] showed that the supplementation of emulsifiers improved the growth performance of broiler chickens by increasing fatty acid digestibility. However, the effects of emulsifiers (in association with low-energy diets) on growth performance, blood chemistry, and fatty acids traits have been rarely thoroughly investigated, even though the interest in using exogenous emulsifiers has increased in the last several decades. Thus, in this study, it could be hypothesized that the utilization of fats can be increased by emulsifiers, which in turn could enhance the growth performance of broilers. In this sense, the current investigation evaluated the influence of emulsifier supplementation into low-energy diets by the attenuation of growth, feed efficiency, and muscle fatty acid profiles in broilers.

## 2. Material and Methods

### 2.1. Experimental Design and Diet Preparation

The study was approved by the Ethics Committee of Local Experimental Animals Care Committee and conducted in accordance with the guidelines of Kafrelsheikh University, Egypt (Number 4/2016 EC). Three hundred one-day-old male birds (Ross 308) were kept in bens and divided randomly into 3 treatments, and each treatment divided into 4 replicates (25 birds/rep). The first treatment was served as control and fed on control diets containing the optimized energy requirements (3000, 3100, and 3180 kcal/kg) for starter, grower, and finisher diets, respectively. The second treatment was fed diets 50 kcal/kg less than control and the third treatment was fed diets 50 kcal/kg less than control and supplemented with 500 g/ton of emulsifiers (Table 1). The emulsifiers used in this study were called Liposorb^®^ and provided from CEVA POLCHEM PVT. LTD., India. Liposorb^®^ contains three types of emulsifiers (Phosphatidyl Choline (PC), Lysophosphatidyl Choline (LPC), PolyEthylene Glycol Ricinoleate (PEGR)), and the optimum dose is 500 g Liposorb®/ton feed. The dose used in the current study was selected based on the study of Bontempo et al. [12] and Zhao et al. [13].

The birds were placed inside a room equipped with 12 pens (3 treatments/4 replicates each) with a chain feeder system and automatic nipple cup drinker in a completely randomized design. Feed and water were provided for ad libitum consumption for 35 days. The light cycle and temperature were the same in the experimental groups. The photoperiod was 24 h of light from day 0 to day 7 and 23 h of light from day 7 to the end of the trial. Room temperature was 25–29 ℃ with proportional moisture between 50% and 70% during the trial. Birds’ body weighing was on record every three days, and feed consumption was on file every day over the empirical period.

### 2.2. Final Sampling

At the end of the trial period, 36 chicks (12 birds/treatment) were randomly chosen, weighed, and gently slaughtered to collect the breast muscle, liver, abdominal fat, and heart organs for offal weight. The blood samples were collected into heparinized test tubes and centrifuged at 5900 × g for 10 min at 4 °C to collect the plasma and finally kept at −20 °C for further analysis. Three days before the end of the trial, 20 birds per group were housed in batteries for digestibility tests where the excreta and feed were collected. Subsequently, the samples were desiccated by the drying kiln at 60 °C for 24 hrs. Then, the dried samples were grinded and kept for protein, lipid, and fiber analysis by following the standard procedure [14]. 

### 2.3. Blood Biochemical Analysis

Plasma triglycerides (TG), total cholesterol (TC), high-density lipoprotein cholesterol (HDL-cholesterol), low-density lipoprotein cholesterol (LDL-cholesterol), glucose, glutamic oxalacetic transaminase (GOT), total protein, albumin, and globulin were tested calorimetrically by using trade kits (Diamond Diagnostics, Egypt) according to the steps outlined by the manufacturer. 

Muscle total lipid content and fatty acid profile analysis were measured using gas-liquid chromatography (GLC) according to the method of Saleh [15]. 

Liver thiobarbituric acid retroactive substances (TBARs) concentration was tested by the process of Goodla et al. [16]. The α-tocopherol concentration in muscle was tested by the HPLC according to the method of Faustman et al. [17]. 

### 2.4. Statistical Analysis

Shapiro–Wilk and Levene’s tests confirmed normal distribution and variance homogeneity. All statistical differences were assessed by one-way analysis of variance tests (SPSS version 17, SPSS Inc., Ill., USA) with Tukey’s multiple test where differences in experimental groups occurred. The level of significance was accepted at *P* < 0.05. All data are presented as means ± standard error (SE).

## 3. Results

### 3.1. Growth Performance and Carcass Composition

Feed intake was not influenced by a low-energy diet supplemented with an emulsifier (*P* = 0.42), however, body weight gain and FCR were enhanced (*P* = 0.005 and *P* = 0.044, respectively) (Table 2). Crude protein utilization and ether extract utilization were decreased by reducing the energy level, but they were increased by emulsifier supplementation (*P* = 0.022 and *P* = 0.011, respectively) (Table 2). However, crude fiber utilization and mortality rate were not influenced by dietary treatments (*P* = 0.32) (Table 2). Carcass % and abdominal fat % were decreased by reduced energy level, but they were increased by adding emulsifier (*P* = 0.045 and *P* = 0.018, respectively). On the other hand, breast and thigh muscles, heart, and liver percentages were not affected by the reduction of energy or addition of emulsifier (*P* > 0.05) (Table 2).

### 3.2. Blood Biochemical Parameters

Plasma total cholesterol, HDL-cholesterol, total protein, and globulin were decreased in the low-energy diet group without emulsifier but they were increased by emulsifier supplementation (*P* = 0.008, *P* = 0.005, *P* = 0.037, and *P* = 0.005, respectively) (Table 3). However, plasma LDL-cholesterol, triglycerides, glucose, GOT, albumin, creatinine, and uric acid were not influenced by dietary treatments (*P* > 0.05) (Table 3). 

### 3.3. Fatty Acid Profiles

Liver TBARs and muscle palmitic acid concentrations were decreased by reducing the energy level and supplementation with emulsifier (*P* = 0.014 and *P* = 0.042, respectively). However, muscle total lipids and α-tocopherol, oleic acid, linoleic acid, and α-linolenic acid were not affected by the tested diets (*P* > 0.05) (Table 4).

## 4. Discussion 

The growth performance was influenced by the reduction of the energy rate in this study. Normally, birds utilize the energy for life maintenance and body building. Subsequently, when birds were fed low-energy diets, the priority of energy utilization can be used for life preservation and in turn, the growth performance might be negatively affected. Following this hypothesis, the results of the present study displayed reduced body weight gain and increased FCR following feeding with a low-energy diet. However, dietary emulsifier presented a practical strategy to increase the growth performance and feed utilization in broilers fed low-energy diets in the present study (high body weight with low FCR). These results are in line with previous studies [11,12,18]. The obtained results revealed that feed intake was not affected by the experimental diets in the present study. Similarly, Kaczmarek et al. [19] reported that dietary emulsifier did not influence the feed intake in broilers. The increased growth performance in the present study might be related to the improvement in crude protein and fat utilization. San Tan et al. [11] and Zhang et al. [18] also illustrated that the inclusion of emulsifiers improved fat digestion and absorption as well as the nutrient digestibility and consequently resulted in enhancing the growth performance in broilers. The emulsifiers are reported to increase the integration of micelles in the gut lumen, which in turn increases the fat digestibility [13]. Thus, it might be hypothesized that the improved digestibility of the fat is an effect of emulsifiers on proteolysis [5,7]. 

The abdominal fat % was significantly decreased by reduced energy level, but it was increased by emulsifier inclusion in the present study. Indeed, a higher rate of dietary metabolizable energy can increase abdominal fat [20,21]. Similarly, Zaman et al. [22] reported that abdominal fat was increased by using diets with high metabolizable energy content. High-energy diets could increase the bulk of fat in broilers’ bodies, and in turn, raises the level of abdominal fat when compared with low-energy diets [23].

By including an emulsifier in low-energy diets, plasma HDL-cholesterol and globulin concentrations were increased, while, plasma LDL-cholesterol, triglycerides, and glucose concentrations were decreased in the current study. Plasma GOT, albumin, creatinine, and uric acid concentrations were not influenced by the tested diets. To the knowledge of the authors, the impact of emulsifier on HDL:LDL levels in the blood plasma of broilers has not been documented yet. Emulsification could reduce the level of free fatty acids and total cholesterol in plasma by lowering the secretion of lipoprotein molecules in the blood [24]. It has been reported that total cholesterol, triglycerides, and HDL-cholesterol of broilers fed feed including plant oil or animal fat were not influenced by dietary emulsifiers [25,26]. On the contrary, broilers fed dietary emulsifier (sodium stearoyl-2-lactylate) displayed low blood triglycerides in comparison with those fed high-energy diets without the addition of emulsifier [27]. 

The discrepancy of the production and removal of oxidants from the organism cells is called oxidative stress [28]. Poor nutritional value of the poultry feeds is among the main reasons for the oxidative stress [29]. However, including feed additives such as antioxidants and emulsifiers in poultry diets has been recognized as an effective strategy to alleviate the impaired effects induced by oxidants on broiler performance [5,30]. TBARs are an indirect marker of oxidative stress, but they are a direct marker of lipid damage caused by increased oxygen under stressful conditions, and α-tocopherol has a crucial role as an antioxidant protecting the lipids from peroxidation [31,32]. In the present study, dietary emulsifiers lowered the TBAR levels in the liver of birds, which confirms that the lipid peroxidation was decreased by feeding emulsifiers [33]. In addition, the level of α-tocopherol is relatively increased by feeding emulsifier, which might be involved in reducing the lipid peroxidation in the liver [34,35]. These results mention that the inclusion of emulsifier in the feed of broilers could enhance the meat quality variables, such as drip loss and tenderness. Outcomes of the current experiment are in accordance with Attia and Kamel [36], who documented that TBARs were reduced by increasing soy lecithin rate in rabbit diets. Such refinements in this study agreed with Al-Daraji et al. [37]. Similarly, emulsifiers have neuroprotective and antioxidative properties, and they diminishes liver damage and enhances oxidative strength [27]. King et al. [38] stated that oxidative stabilization might be related to the ability of phospholipids to pool a hydrogen atom from the amino group that moves the oxidized phenolic molecule of the real antioxidant. Moreover, Judde et al. [39] elucidated that the antioxidative properties of emulsifiers depend on fatty acid structure and tocopherol concentration. 

The muscle palmitic acid content as saturated fatty acids was decreased by reduced energy and emulsifier, however, muscle oleic acid, linoleic acid, and α-linolenic acid as unsaturated fatty acids were not significantly affected in this study. The emulsifiers can aid to fix the free fatty acids that are seldom soluble by themselves in bile salt micelles, and in this way they raise the digestibility of saturated fatty acids [25]. 

In the present study, we found that the inclusion of exogenous emulsifiers in broilers’ diets with low energy clearly enhanced the weight gain and reduced the feed conversion ratio. Further, the protein and lipid utilization was increased by emulsifier supplementation. Interestingly, the liver TBARs and muscle palmitic acid concentrations were decreased by reducing energy level and supplementation with emulsifier, which confirms the protective role of emulsifiers against oxidative stress. The plasma total cholesterol, HDL-cholesterol, total protein, and globulin were also increased by emulsifier supplementation. The obtained results are in agreement with previous studies investigating the importance of using emulsifiers in poultry diets [12,40,41,42,43]. However, further experiments are needed by using different alternative fat sources with different emulsifier additions and doses to explore their effects on the fatty acid profiles and the performances of broilers in different periods of age.

## 5. Conclusions 

It could be concluded that dietary supplementation of a mixture of emulsifiers in low-energy diets exhibited similar or more effective effects on growth performance, nutrient utilization, lipid peroxidation, and modified plasma lipids than the high-metabolizable energy diet in broiler chickens. Due to the continuous rise of ingredient price and energy cost, the obtained results confirm the concept of using emulsifiers in low-energy diets to reduce the cost of poultry feeding and in turn increase the profitability as well as reduce the price of the product, leading improved welfare of mankind. However, future studies are required to clarify the mechanistic role of emulsifiers in improving the performances of broiler feed diets with low energy using advanced molecular tools. 

## Figures and Tables

**Table 1 animals-10-00437-t001:** Composition of the experimental diets.

Ingredient	Basal Diet			Low Energy Diet (−50 kcal)			Low Energy Diet (−50 kcal) + Emulsifier		
	Starter	Grower	Finisher	Starter	Grower	Finisher	Starter	Grower	Finisher
Yellow Corn	559.5	601.5	659.5	570.5	612.5	670.5	570	612	670
Soybean meal (44%)	325	276	215	324	275	214	324	275	214
Soy oil	15	22	25	5	12	15	5	12	15
Toxin binders	0.5	0.5	0.5	0.5	0.5	0.5	0.5	0.5	0.5
Broiler concentrate (45%) ^1^ + Premix ^2^	100	100	100	100	100	100	100	100	100
Emulsifiers	0	0	0	0	0	0	0.5	0.5	0.5
Total	1000	1000	1000	1000	1000	1000	1000	1000	1000
Calculated analysis									
CP (%)	23.09	21.26	19.02	23.09	21.26	19.0.	23.085	21.26	19.02
ME (kcal/Kg)	3003	3093	3177	2955	3045	3128	2954	3043	3127
Ca (%)	1.01	0.98	0.88	1.01	0.99	0.88	1.01	0.98	0.88
Lysine (%)	1.352	1.217	1.152	1.349	1.215	1.149	1.35	1.216	1.15
Methionine (%)	0.57	0.55	0.52	0.57	0.55	0.52	0.57	0.55	0.52

^1^ Broiler Concentrate—crude protein, 45%; ME, 2800 kacl/kg; calcium, 9%; phosphorus, 2.74%; Na, 1.9%; CL, 1.5%; lysine, 2.8%; methionine, 2.5%. ^2^ Premix (g/kg feed)—calcium hydrogen phosphate, 20; calcium carbonate, 8.15; sodium chloride, 5.1; (µg/kg feed): iron sulfate, 400; copper sulfate, 31.5; zinc sulfate, 176; manganese sulfate, 152; sodium iodate, 0.55; sodium selenite, 0.27. Retinol, 1.4; DL-α-tocopherol acetate, 6.5; thiamine hydrochloride, 2.6; riboflavin, 6.5; pyridoxine hydrochloride, 1.30; calcium D-pantothenate, 10.4; nicotinic acid, 26; (µg/kg feed): menadione sodium bisulfite, 650; D-biotin, 70; choline chloride, 780; pteroylglutamic acid, 520; cyanocobalamin, 26; cholecalciferol, 13.

**Table 2 animals-10-00437-t002:** Effect of test diets on growth performance, nutrient utilization, and organ weights in broilers.

Item	Treatment			*P*-Value
	Basal Diet	Low Energy Diet (−50 kcal)	Low Energy Diet (−50 kcal) + Emulsifier	
Initial body weight (g)	42.23 ± 0.14	42.10 ± 0.23	42.13 ± 0.24	0.900
Body weight gain (g/34 day)	1769 ± 10.00 ^a^	1624 ± 41.00 ^b^	1761 ± 12.00 ^a^	0.005
Feed intake (g/34 day)	3043 ± 80.00	2883 ± 85.00	2994 ± 86.00	0.420
Feed conversion ratio (g gain/g feed)	1.68 ± 0.05^b^	1.73 ± 0.06 ^a^	1.66 ± 0.06 ^b^	0.044
Crude protein utilization (%)	68.00 ± 0.50 ^a^	64.30 ± 1.10 ^b^	67.00 ± 0.70 ^a^	0.022
Crude fiber utilization (%)	34.50 ± 0.70	33.00 ± 0.60	34.30 ± 0.90	0.320
Ether extract utilization (%)	46.75 ± 1.50 ^a^	42.00 ± 0.50 ^b^	45.75 ± 0.60 ^a^	0.011
Mortality (%)	2.00 ± 1.15	3.00 ± 1.00	2.00 ± 1.15	0.767
Organ weight (% body weight)				
Carcass	68.86 ± 0.80 ^a^	66.11 ± 1.30 ^b^	68.39 ± 0.60 ^a^	0.045
Breast muscle	23.46 ± 0.38	22.23 ± 0.93	23 ± 0.22	0.360
Thigh muscle	19.65 ± 0.50	18.78 ± 0.90	19.4 ± 0.50	0.630
Abdominal fat	1.88 ± 0.15 ^a^	1.47 ± 0.05 ^b^	1.77 ± 0.05 ^a^	0.018
Liver	2.47 ± 0.08	2.44 ± 0.07	2.48 ± 0.06	0.901
Heart	0.81 ± 0.02	0.85 ± 0.02	0.8 ± 0.02	0.122

^a^^,b^ Values expressed as means ± SE. Means within the same row with different superscripts differ (*P* < 0.05).

**Table 3 animals-10-00437-t003:** Effect of test diets on blood plasma parameters.

Item	Treatment			*P*-Value
	Basal Diet	Low-Energy Diet (−50 kcal)	Low-Energy Diet (−50 kcal) + Emulsifier	
Triglycerides (mg/ml)	102.00 ± 4.20	97.80 ± 4.60	99.13 ± 3.30	0.752
Total cholesterol (mg/ml)	146.00 ± 2.50 ^a^	131.00 ± 3.40^b^	134.00 ± 3.80 ^b^	0.008
HDL- cholesterol (mg/ml)^1^	27.25 ± 0.31 ^a^	24.00 ± 0.87 ^b^	25.75 ± 0.56 ^ab^	0.005
LDL- cholesterol (mg/ml)^2^	109.00 ± 3.70	103.00 ± 2.40	101.00 ± 2.10	0.137
Glucose (mg/ml)	180.40 ± 8.50	177.00 ± 9.30	175.00 ± 7.40	0.891
GOT (U/I)^3^	253.00 ± 9.90	241.00 ± 8.30	245.00 ± 10.00	0.667
Total protein (mg/dL)	4.20 ± 0.10 ^a^	3.75 ± 0.08 ^b^	4.06 ± 0.16 ^ab^	0.037
Albumin (mg/dL)	2.23 ± 0.04	2.22 ± 0.04	2.20 ± 0.07	0.933
Globulin (mg/dL)	1.91 ± 0.08 ^a^	1.48 ± 0.06 ^b^	1.81 ± 0.11 ^a^	0.005
Creatinine (mg/ml)	0.625 ± 0.05	0.55 ± 0.03	0.59 ± 0.03	0.365
Uric acid (mg/ml)	5.40 ± 0.30	5.70 ± 0.40	5.75 ± 0.31	0.672

^a,b^ Values expressed as means ± SE. Means within the same row with different superscripts differ (*P* < 0.05). ^1^ HDL, high-density lipoprotein; ^2^ LDL, low-density lipoprotein; ^3^ GOT, glutamic oxaloacetic transaminase.

**Table 4 animals-10-00437-t004:** Effect of test diets on liver thiobarbituric acid retroactive substances (TBARs), muscle total lipid, α-tocopherol, and fatty acid profile contents in muscle.

Item	Treatment			*P*-Value
	Basal Diet	Low Energy Diet (−50 kcal)	Low Energy Diet (−50 kcal) + Emulsifier	
Liver				
TBARs (nmol MDA/g)	3.60 ± 0.08 ^a^	2.85 ± 0.22 ^b^	2.73 ± 0.2 ^b^	0.014
Muscle				
Total lipid (%)	3.03 ± 0.11	2.98 ± 0.08	2.93 ± 0.13	0.812
α-Tocopherol (mg/100 g)	0.21 ± 0.01	0.20 ± 0.01	0.22 ± 0.02	0.690
Fatty acid (mg/100 mg fat)				
Palmitic acid (16:00)	1.36 ± 0.04 ^a^	1.23 ± 0.03 ^b^	1.19 ± 0.04 ^b^	0.042
Oleic acid [18:1 (n-9)]	1.07 ± 0.06	1.07 ± 0.02	1.13 ± 0.01	0.520
Linoleic acid [18:2 (n-6)]	1.88 ± 0.23	1.88 ± 0.21	1.92 ± 0.23	0.986
Omega-3 [18:3 (n-3)]	0.06 ± 0.02	0.05 ± 0.02	0.07 ± 0.01	0.731

^a,b^ Values expressed as means ± SE. Means within the same row with different superscripts differ (*P* < 0.05).

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
