# Peer review of "A Mixture of Exogenous Emulsifiers Increased the Acceptance of Broilers to Low Energy Diets: Growth Performance, Blood Chemistry, and Fatty Acids Traits"

_animals, 2020, doi:10.3390/ani10030437_

Round 1
Reviewer 1 Report
In abstract authors should give P values.
Authors must provide code of Ethical committee approval.
Statistical analysis description must be improved. What test did authors use for confirmation of normality of distribution of their data? please explain or rephrase term "General Linear ideal".
Though the whole manuscript authors should give exact P values and state exact statistical test that was used for analysis.
The number of decimal places must be equal in all numbers.
Authors should emphasize strengths and limitations of the study in the last paragraph of discussion section.
Conclusion should be expanded.
List of references should be updated with the newest ones. References must be listed according to the guidelines.
Author Response
Responses to the comments of Reviewer #1
With all due respect to the reviewer, your revisions and suggestions enabled us to improve the paper quality. In accordance with your wishes, we have now changed this manuscript to be more suitable for publication in animals. In the following are our point-by-point responses to each of your own comments:
- In abstract authors should give P values.
- Authors must provide code of Ethical committee approval.
- Statistical analysis description must be improved. What test did authors use for confirmation of normality of distribution of their data? please explain or rephrase term "General Linear ideal".
- Though the whole manuscript authors should give exact P values and state exact statistical test that was used for analysis.
- The number of decimal places must be equal in all numbers.
- Authors should emphasize strengths and limitations of the study in the last paragraph of discussion section.
- Conclusion should be expanded.
- List of references should be updated with the newest ones. References must be listed according to the guidelines.
- Response: Thanks for your comments and revisions. The manuscript has been revised carefully in the light of your comments. Kindly check the revised version of the manuscript.

Reviewer 2 Report
In this manuscript, the authors have presented the results of their studies on effects of emulsifiers supplemented with low energy diets on the growth performance, blood Chemistry, and fatty acid status of broiler chickens.
The presented data are sound. However, there are a few points that need to be addressed.
- Authors should specify the kind/type/name of the emulsifiers used in the title.
- The aim of the study is not clear.
- Authors are suggested to specify the difference between Bontempo et al. (12) study to their current study.
- Authors are suggested to include the following references at line no. 46 after broiler chickens.
- A K Panda et al., 2015. Growth performance, carcass characteristics, fatty acid composition and sensory attributes of meat of broiler chickens fed diet incorporated with linseed oil. Indian Journal of Animal Sciences. 85 (12): 1354-1357.
- A K Panda et al., 2016. Effect of Dietary Incorporation of Fish Oil on Performance, Carcass Characteristics, Meat Fatty Acid Profile and Sensory Attributes of Meat in Broiler Chickens. Animal Nutrition and Feed Technology. 16 (3): 417-425.
- In experimental design and diet preparation section authors have stated that the birds were kept in bens and also stated that the trial was conducted in open poultry farm (line no.81)? Authors should clearly mention the method or way they have conducted the trial? How the authors demarcated birds of one group to the other if the trail was in open poultry farm?
- Which sex of birds was employed for the experiment? Animal ethics approval number should be provided.
- On what basis the dosage of emulsifier was determined for the in vivo supplementation?
- Why only liver tissue was selected for TBARS analysis?
- Why authors have not considered other organs such as gizzard, bursa, and spleen to determine the growth performance?
- Should represent the names of the fatty acids analyzed in the study in table 4.
- HI titer and organoleptic analysis could be done.
- Authors are suggested to include recent references in the manuscript instead of old references.
- Does the deviated values (± values) in the table are standard errors or standard deviations (SE/SD)? Should be represented in table footnotes.
- Authors should discuss the role lipid peroxidation and compare their results with others work in discussion part.
- Authors are suggested to include the following reference at line 111 instead of reference no.15.
Lavanya et al., 2019. Protective effects of Ammannia baccifera against CCl4-induced oxidative stress in rats. International journal of environmental research and public health. 16 (8), 1440.
- Authors should check the conclusion part: Are the results really supporting the conclusion? The tested /selected parameters are sufficient for the conclusion? Should re-write the conclusion by including future directions.
- References should be cited by following journal style/format.
- Need to check for typographical errors, plagiarism, punctuation, and grammar throughout the manuscript.
Author Response
Responses to the comments of Reviewer #2
With all due respect to the reviewer, your revisions and suggestions enabled us to improve the paper quality. In accordance with your wishes, we have now changed this manuscript to be more suitable for publication in animals. In the following are our point-by-point responses to each of your own comments:
- Authors should specify the kind/type/name of the emulsifiers used in the title.
- Response: Thanks for your comment. Due to the large name of the mixture of emulsifiers used in the current study, we changed the title as following and detailed the name of emulsifiers in the abstract and text.
A Mixture of Exogenous Emulsifiers Increased the Acceptance of Broilers to Low Energy Diets: Growth Performance, Blood Chemistry, and Fatty Acids Traits
- The aim of the study is not clear.
- Authors are suggested to specify the difference between Bontempo et al. (12) study to their current study.
- Authors are suggested to include the following references at line no. 46 after broiler chickens.
A K Panda et al., 2015. Growth performance, carcass characteristics, fatty acid composition and sensory attributes of meat of broiler chickens fed diet incorporated with linseed oil. Indian Journal of Animal Sciences. 85 (12): 1354-1357.
A K Panda et al., 2016. Effect of Dietary Incorporation of Fish Oil on Performance, Carcass Characteristics, Meat Fatty Acid Profile and Sensory Attributes of Meat in Broiler Chickens. Animal Nutrition and Feed Technology. 16 (3): 417-425.
- In experimental design and diet preparation section authors have stated that the birds were kept in bens and also stated that the trial was conducted in open poultry farm (line no.81)? Authors should clearly mention the method or way they have conducted the trial? How the authors demarcated birds of one group to the other if the trail was in open poultry farm?
- Which sex of birds was employed for the experiment? Animal ethics approval number should be provided.
- On what basis the dosage of emulsifier was determined for the in vivo supplementation?
- Response: Thanks for your comment. Kindly, check the revised manuscript. The discussion is greatly improved as per your suggestion.
- Why only liver tissue was selected for TBARS analysis?
- Response: Thanks for your comment. As mentioned in the revised manuscript that is the liver is considered the main organ which clearly can be affected by oxidation. However, we will consider your suggestion in the future studies.
- Why authors have not considered other organs such as gizzard, bursa, and spleen to determine the growth performance?
- Response: Thanks for your comment. In the current study we just focused on the organs which is related to the fat deposition and fattening. However, we will consider your suggestion in the future studies.
- Should represent the names of the fatty acids analyzed in the study in table 4.
- HI titer and organoleptic analysis could be done.
- Response: Thanks for your comment. However, we will consider your suggestion in the future studies.
- Authors are suggested to include recent references in the manuscript instead of old references.
- Does the deviated values (± values) in the table are standard errors or standard deviations (SE/SD)? Should be represented in table footnotes.
- Authors should discuss the role lipid peroxidation and compare their results with others work in discussion part.
- Authors are suggested to include the following reference at line 111 instead of reference no.15.
Lavanya et al., 2019. Protective effects of Ammannia baccifera against CCl4-induced oxidative stress in rats. International journal of environmental research and public health. 16 (8), 1440.
- Authors should check the conclusion part: Are the results really supporting the conclusion? The tested /selected parameters are sufficient for the conclusion? Should re-write the conclusion by including future directions.
- References should be cited by following journal style/format.
- Need to check for typographical errors, plagiarism, punctuation, and grammar throughout the manuscript.
- Response: Thanks for your comment. Kindly, check the revised manuscript. All done. The discussion and conclusion are greatly improved as per your suggestion.
Thank you once again for your valuable comments and suggestions. We are hopeful that our revised focus helps to improve your opinion of our work.

Round 2
Reviewer 2 Report
The revised version of this manuscript was improved. Still few points need to be considered.
Authors need to specify the significance of the study for the mankind.
Once again need to check for the reference format/style, both in text and reference section.
Need to check for any grammatical mistakes, typographical errors and plagiarism once again.
Author Response
With all due respect to the reviewer, your revisions and suggestions enabled us to improve the paper quality. In accordance with your wishes, we have now changed this manuscript to be more suitable for publication in animals. In the following are our point-by-point responses to each of your own comments:
Comment: The revised version of this manuscript was improved. Still few points need to be considered. Authors need to specify the significance of the study for the mankind.
Response: At the of ABSTRACT and CONCLUSION, we added the significance of obtained results for the mankind.
Lines 41-43: “Getting such benefits in broilers is a necessity to reduce the feed cost and consequently the price of the product, which will lead to improve the welfare of the mankind.”
Lines 264-267: “Due to the continuous rise of ingredient price and energy cost, the obtained results confirm the concept of using emulsifiers in low energy diets to reduce the cost of poultry feeding and in turn increase the profitability as well as reduce the price of the product, leading to improve the welfare of the mankind.”
Comment: Once again need to check for the reference format/style, both in text and reference section.
Response: The reference format/style, both in text (Line 92) and reference section was checked and corrected. We corrected in REFERENCES # 1, 7, 15, 16, 20, 23, 25, 30, 33, 37 & 43.
Comment: Need to check for any grammatical mistakes, typographical errors and plagiarism once again.
Response: Grammar, typographical errors and plagiarism were reviewed and corrected by English expert. Also, punctuation was corrected in the whole manuscript.